# Coproduction of accessible digital mental health supports in partnership with young people from marginalised backgrounds: a scoping review protocol

Carmen Kealy [iD],[1] Courtney Potts [iD],[2] Maurice D Mulvenna [iD],[3] Gary Donohoe [iD],[4] Siobhan O'Neill [iD],[2] Margaret M Barry [iD] [1]

¹Health Promotion Research Centre, University of Galway, Galway, Ireland
²School of Psychology, Ulster University, Coleraine, UK
³School of Computing, Ulster University, Belfast, UK
⁴School of Psychology, University of Galway, Galway, Ireland

**Correspondence to**
Courtney Potts;
c.potts@ulster.ac.uk

## ABSTRACT

**Introduction** Despite the evidence supporting the value of digital supports for enhancing youth mental health services, there is a lack of guidance on how best to engage with young people in coproduction processes during the design and evaluation of these technologies. User input is crucial in digital mental health, especially for disadvantaged, vulnerable and marginalised young people as they are often excluded from coproduction. A scoping review of international literature written in English will explore the coproduction processes with marginalised young people in digital mental health supports, from mental health promotion to targeted interventions. The review is guided by the research question: what are the most appropriate coproduction processes for engaging young people, especially marginalised young people, in the different stages of designing and evaluating digital mental health supports? The review aims to map and summarise the evidence, inform the overall research project and address the knowledge gaps.

**Methods and analysis** The scoping review uses Arksey and O'Malley's framework and the Preferred Reporting Items for Systematic Reviews and Meta-Analysis Protocols Extension for Scoping Reviews. From 22–24 October 2023, PubMed, Scopus, EBSCO, ASSIA, Web of Science, Ovid MEDLINE, Cochrane database, Embase, Google Scholar, ProQuest, OAlster and BASE will be systematically searched. Papers from 2021 onwards with a range of study designs and evidence that illustrate engagement with marginalised young people (aged 16–25) in the design, implementation and evaluation of digital technologies for young people's mental health will be considered for inclusion. At least two reviewers will screen full texts and chart data. The results of this review will be summarised quantitatively through numerical counts of included literature and qualitatively through a narrative synthesis.

**Ethics and dissemination** Ethical approval is not required. Results will be disseminated through publications in peer-reviewed journals.

**Trial registration number** This scoping review protocol has been registered with the Open Science Framework (https://osf.io/9xhgv).

### STRENGTHS AND LIMITATIONS OF THIS STUDY

⇒ This review will explore the literature on coproduction across all stages of design and evaluation of digital mental health supports for young people, with a particular emphasis on marginalised young people who are often excluded from these processes.

⇒ The review will be limited to studies published from 2021 onwards to capture the most current literature on the use of coproduction methods and the dynamic nature of this topic.

⇒ Due to the scoping review design, quality assessments of the included articles will not be performed.

⇒ The synthesis of knowledge will serve as a basis for the wider research project and concrete calls to action.

the accessibility of mental health services and support for young people, including marginalised young people with mental health needs who would otherwise find it difficult to seek help.[1] Guidelines with regard to digital mental health interventions highlight the importance of early user input in the development, implementation and evaluation of technologies to ensure that they are engaging, feasible, acceptable and potentially effective.[2] However, there is a lack of guidance concerning the most appropriate coproduction processes for engaging young people in the different stages of assessing digital mental health technologies, especially with regard to those who are disadvantaged, vulnerable and/or marginalised.[3–5]

Focusing on the innovative digital mental health approaches led by young people, this review will consider coproduction in the design, development, implementation and evaluation phases of digital mental health supports for marginalised young people, as well as overall evaluations of coproduction processes in this context. The review is part of a larger project called 'Atlantic Futures'.[6]

## INTRODUCTION
There is increasing evidence that some digital mental health supports may improve

One of the work packages (Research Stream 4) within this project explores the accessible digital supports and blended services to promote the mental health of youth on a shared island basis in Ireland and Northern Ireland. This is to be achieved through engaging with young people (aged 16–25), including those who are marginalised, and the staff who support them in health, youth and community services.

## Marginalisation

This scoping review focuses on young people categorised as 'marginalised' in the digital mental health support context. A briefing paper produced by UNICEF's Office of Research defines 'disadvantaged, vulnerable and/ or marginalised adolescents as individuals aged 10–19, who are excluded from social, economic and/or educational opportunities enjoyed by other adolescents in their community due to numerous factors beyond their control'.[7] These factors include those operating at the social level (such as economic inequality, violence, stigma, racism and migration), family level (including neglect and abuse) and individual level (eg, disability and ethnicity). Examples of disadvantaged, vulnerable and/or marginalised young people include immigrants, refugees, orphans and those who belong to stigmatised indigenous, ethnic or religious groups. They also include individuals who belong to sexual minorities (eg, gay, lesbian, bisexual and queer) or gender minorities (eg, transgender and gender diverse).

Previous reviews on coproduction in digital mental health interventions have highlighted either an under-representation of young people with learning disabilities and specific difficulties[3] or a marked absence of young people including, but not limited to those who are migrants, asylum seekers and refugees, those experiencing homelessness but also those from socioeconomically deprived backgrounds.[8]

In the context of digital mental health technologies, the concept of digital marginalisation is also highly relevant, and this typically refers to people who do not have or use the internet, who lack access to fast and reliable internet connection (eg, people living in rural areas) and who lack the skills or access to developing skills for using the internet.[8] Many marginalised groups face barriers in accessing the online services, which can further perpetuate the 'digital divide', a technology-based form of social inequality that can increase marginalisation in other areas of their lives.[9]

Considering the above, the project seeks to take into account the specific needs of young people who are disadvantaged, vulnerable and/or marginalised from a socioeconomic, cultural and digital perspective.

Thus, in order to investigate coproduction and marginalisation, the scoping review will include, but not be limited to, the following groups of young people who have been identified as marginalised in the Irish context
► Living in isolated rural areas[10]
► Socioeconomically deprived
► Unemployed
► Not in education, employment or training
► Immigrants
► Asylum seekers
► Refugees
► Travelling community
► Lesbian, gay, bisexual, transgender, and queer (LGBTQ+) community
► Living with a disability
► Homeless

This scoping review intends to explore the most appropriate coproduction processes for engaging young people in the different stages of designing, developing, implementing and evaluating digital mental health technologies, especially with regard to those who are marginalised. Due to the exploratory nature of this research question, a scoping review approach has been selected to explore the breadth and depth of the extant literature on coproduction processes with marginalised young people in the context of digital mental health supports, ranging from mental health promotion and primary prevention to targeted interventions. This includes all types of digital interventions, such as websites and apps but not studies that report on digital delivery of services that are typically administered face-to-face such as remote counselling. The review aims to map and summarise the evidence, inform the overall research project, as well as identify and address the knowledge gaps.[11]

## METHODS AND ANALYSIS

This scoping review is prepared in accordance with the Preferred Reporting Items for Systematic Reviews and Meta-Analysis Protocols Extension for Scoping Reviews guidelines[12] (online supplemental file 1). A scoping review is a systematic way of exploring a topic area, in which the main concepts and knowledge gaps are identified within a developing field of research.[13] In contrast to systematic reviews, the research questions for a scoping review are frequently broader and more exploratory,[13] and hence appropriate for this topic due to insufficient research and a lack of synthesised knowledge. Guided by Arksey and O'Malley's[13] five-step set, but also Levac and colleagues'[14] update of this framework, the review process will follow this tentative timeline:
1. Identifying the research question— October 2023
2. Identifying relevant studies— October 2023
3. Study selection—November–December 2023
4. Charting the data—January 2024
5. Collating, summarising and reporting the results— May 2024

## Identifying the research question

In keeping with the exploratory nature of scoping reviews, the research team identified one main research question: what are the most appropriate coproduction processes for engaging young people in the different stages of designing and assessing digital mental health

technologies, especially regarding those who are marginalised?

## Identifying relevant studies

Following a preliminary search to see the volume returned, the following electronic databases will be systematically searched for international literature published in English: PubMed, Scopus, EBSCO (CINAHL/PsycINFO/PsycArticles), Applied Social Sciences Index & Abstracts, Web of Science, Ovid MEDLINE, Cochrane database, Embase, Google Scholar, ProQuest Dissertations & Theses A&I, OAIster (Catalog of open access resources) and Bielefeld Academic Search Engine (BASE). The focus of this scoping review is informed by previous reviews on coproduction and/or marginalised young people in digital mental health interventions,[1–5] which have been conducted and/or published since the onset of the global coronavirus pandemic and associated restrictions in March 2020. Highlighting an under-representation of marginalised young people, searches will lead on from these reviews and be limited from January 2021 to the date of search commencement in October 2023. This ensures that the review reflects the most current literature on the use of coproduction methods and thus the nature of this dynamic research topic. Published work from 2021 onwards is included to mirror the current state of literature more accurately, given the accelerated use and assessment of digital mental health technologies throughout the sector during the pandemic.[15]

Title, abstract and keyword fields will be searched with the following search terms (table 1) where advanced search options are possible.

For Google Scholar and BASE, the following search terms will be used: (Co-production OR Co-design) AND (Digital mental health) AND (marginal*youth* OR disadvant* adolescen* OR vulnerable teen* OR support* OR intervention OR promot* OR prevent* OR develop* OR evalua* OR implem*).

An example of the electronic search strategy for PubMed has been included (online supplemental file 2).

Results from academic and grey literature databases will be downloaded in Research Information Systems (RIS) format and uploaded to the Covidence software,[16] a screening and data extraction tool for conducting reviews, which allows screening to be more efficient and easily tracked.

## Study selection

Two senior research team members will be assigned to review all the articles identified from the databases independently, and 20% of the overall article returns will also be screened by two other members of the research team. Titles and abstracts will be reviewed based on the initial search and will be included in accordance with the eligibility criteria described below. Duplicate articles will be removed, while full texts will be examined to create a final list of the included studies. Any disagreements on the inclusion of any articles will be resolved through discussion and consultation with other team members if necessary.

The reference lists of the included papers will be examined as well to verify that all the relevant publications are incorporated. As the review develops, other techniques for searching may be implemented and any additions or changes will be noted.

## Eligibility criteria

Empirical research and discussions in the literature will be included in the review if they: (1) are published since January 2021; (2) are published in English; (3) focus on coproduction/codesign of digital youth mental health supports with marginalised young people aged 16–25 and (4) use innovative youth-led methodologies to design, develop, implement and/or evaluate digital mental health supports for young people.

| Table 1 | Search terms |
|---|---|
| Mental health | "well-being" OR wellbeing OR stress OR "mental disorder" OR "mental illness" OR "mental health" OR depress* OR "psychological health" OR anxiety OR psychiatric OR "mood disorder" OR "mental disease" |
| Young people | youth* OR young* OR child* OR adolescen* OR student* OR teen* |
| Marginalisation* | marginali* OR disadvan* OR vulnerab* OR depriv* OR "ethnic minorit*" OR immigra* OR homeless OR minorit* OR "low-income" OR disabili* OR isolat* OR LGBTQ+ OR NEET OR NEETs OR refugee OR indigenous OR neurodiver* |
| Interventions | Intervention OR promo* OR prevent* OR program* OR support OR polic* OR implementation OR evaluation OR therap* OR develop* |
| Technology | digital* OR mHealth OR eHealth OR "web-based" OR "internet-based" OR "mobile phone" OR "text message" OR "text-based" OR SMS OR app OR "artificial intelligence" OR tele* OR computeri* OR online OR "electronic health" OR "telemedicine" |
| Co-production | "co-produ*" OR "co-design*" OR "youth-led" OR participatory OR collab* |

*Search terms for the category marginalisation consist of definition terms provided by UNICEF's briefing paper[9] as well as search terms found adequate to capture the groups of young people identified as marginalised in the Irish but also international context.
NEET, not in education, employment or training.

Articles will not be included if they are: (1) non-digital interventions; (2) technologies developed primarily for physical health or diagnostic, screening, monitoring, communication or data management tools; (3) focus on digital mental health supports without young people's input in the development, implementation and/or evaluation of these supports and (4) involving children (under the age of 16) and adults (over the age of 25).

### Charting the data

Data from the identified studies will be collected and charted according to key groupings, for example, based on the focus of the article—design, development, implementation and evaluation. From each eligible article, the reviewers will include authorship, year and journal of publication, the general characteristics of participants (ie, age, gender and sociodemographic information), geographic location of study, study methods, the identified coproduction processes for digital mental health supports, limitations and any other key findings relating to the research question. An example of a charting template has been included (online supplemental file 3). A combination of Excel and Covidence management software will be used to collate the charted data and manage the screening process.

### Collating, summarising and reporting the results

As this scoping review is intended to provide an overview of the existing literature and not to critically appraise the included articles, the risk of bias will not be assessed. However, we will include a discussion of any limitations found in the included body of evidence. The results of this scoping review will be summarised quantitatively by highlighting the amount and type of studies reviewed, and qualitatively through a narrative synthesis. Tables and/or charts will be used to map the study findings and provide an overview, and the screening process will be visualised through a flowchart. Given the exploratory nature of this study, any factors related to coproducing digital mental health supports will be considered for review. Of particular interest is the sourcing of state-of-the-art examples that engage with marginalised young people (aged 16–25 and capturing the out-of-school population of young people in Ireland as per education policies) in an innovative way, not only in the initial exploration/design but also in the implementation and evaluation phase of digital technologies for young people's mental health. The review also seeks to identify published works that have evaluated overall co-production processes in the creation of digital mental health supports for young people.

### Patient and public involvement

The design of this scoping review protocol did not involve patients or the public.

### Ethics and dissemination

As this scoping review is intended to synthesise the current breadth of knowledge on coproduction processes with marginalised youth in the context of digital mental health supports, no ethical approval is required. The results of this scoping review will be disseminated through publications in peer-reviewed journals.

## DISCUSSION

The results of this study will establish what is currently known about the most appropriate coproduction processes for engaging young people in the different stages of assessing digital technologies, especially with regard to those who are disadvantaged, vulnerable and/or marginalised. To the best of our knowledge, this scoping review protocol is the first on this topic and will identify the key themes and gaps in understanding how to engage with marginalised young people (aged 16–25) in an innovative and meaningful way when designing/developing, as well as implementing and evaluating, digital mental health technologies for young people's mental health. The results will serve as the basis for the wider project, including a qualitative exploration of digital mental health challenges facing young people across the island of Ireland; identifying, with young people, the appropriate digital mental health apps and interventions; validating, in partnership with young people, what digital mental health interventions work and using these findings to inform policy and practice recommendations for health providers across the island of Ireland and beyond. The review will contribute to the existing literature in the field by not only highlighting what has been achieved to date but also by proposing potential ways of advancing the inclusion of marginalised young people in the coproduction of digital mental health supports. This will ensure user input, access and subsequent improvement in digital mental health technologies.

X Carmen Kealy @@carmenkealy1?lang=en

**Contributors** CK, CP, MM, GD, SO and MMB contributed to the development of this manuscript. CK conceptualised the research question, designed the study and prepared the first draft of the manuscript. MMB helped refine the research question and provided review expertise. All authors contributed to the refining of the study design as well as to the editing and revising of this protocol. All authors have approved the final manuscript for submission.

**Funding** Atlantic Futures is a €4 million 4-year cross-border research project funded by The North–South Research Programme (422560). The North–South Research Programme is a collaborative scheme funded through the Government's Shared Island Fund. It is administered by the Higher Education Authority on behalf of the Department of Further and Higher Education, Research, Innovation and Science.

**Competing interests** None declared.

**Patient and public involvement** Patients and/or the public were not involved in the design, conduct, reporting or dissemination plans of this research.

**Patient consent for publication** Not applicable.

**Provenance and peer review** Not commissioned; externally peer reviewed.

terminology, drug names and drug dosages), and is not responsible for any error and/or omissions arising from translation and adaptation or otherwise.

**Open access** This is an open access article distributed in accordance with the Creative Commons Attribution 4.0 Unported (CC BY 4.0) license, which permits others to copy, redistribute, remix, transform and build upon this work for any purpose, provided the original work is properly cited, a link to the licence is given, and indication of whether changes were made. See: https://creativecommons.org/licenses/by/4.0/.

**ORCID iDs**
Carmen Kealy http://orcid.org/0000-0001-6373-6494
Courtney Potts http://orcid.org/0000-0002-5621-1611
Maurice D Mulvenna http://orcid.org/0000-0002-1554-0785
Gary Donohoe http://orcid.org/0000-0003-3037-7426
Siobhan O'Neill http://orcid.org/0000-0002-8786-2118
Margaret M Barry http://orcid.org/0000-0002-9464-7176

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
