## [Reviewer comments · BMJ Open]

ARTICLE DETAILS

TITLE (PROVISIONAL)	Co-production of accessible digital mental health supports in partnership with young people from marginalised backgrounds: a scoping review protocol
AUTHORS	Kealy, Carmen; Potts, Courtney; Mulvenna, Maurice; Donohoe, Gary; O'Neill, Siobhan; Barry, Margaret M.

VERSION 1 – REVIEW

REVIEWER	Robards, Fiona University of Sydney, General Practice
REVIEW RETURNED	12-Dec-2023

GENERAL COMMENTS	Thank you for the opportunity to review 'coproduction of accessible digital mental health support in partnership with young people from marginalised backgrounds: a scoping review per protocol'. General comments The topic for this review is interesting and would provide a useful addition to the literature. I realise that youth is increasingly being used as a noun, such as in the expression marginalised youth. However, if you consult the dictionary, youth describes a period of time, not a group of people. I therefore advocate that youth is used as an adjective, not a noun. For example, 'marginalised young people' is much better than 'marginalised youth'. Abstract The abstract reads well. My only suggestion is to see the research question clearly articulated within the abstract. Article summary The strengths and limitations section articulates that there may be a shortage of articles. I noticed that the search strategy calls for the intersection of six groups, and the requirement of articles meeting so many may limit the search. Further, the publication period is very limited. Can these limitations be addressed? A strong limitation is the omission of indigenous young people in the list of marginalised groups and from the search terms. This issue should be addressed. Introduction
--

	Page 4, line 6 The first line of the introduction needs reconsideration. I suggest softening the language to include words such as 'some' and 'may' to make to improve the accuracy of the claim. The first paragraph has both expressions 'with regards to', and 'with regard to'. I suggest editing for consistency. Page 4, line 17 It would be good to define how digital mental health support is conceptualised. Does this include prevention through to treatment? Apps and online programs and telehealth? Page 4, line 31 While it is your choice, I would advise against using an acronym such as DVM. Later in the article (for example, in the research question), you refer to marginalised young people, which is much better. Page 4, line 41 The example groups of marginalised young people must be grammatically accurate and described more respectfully. See the last sentence in the paragraph. It is not appropriate to refer to a group of young people as 'sexual minorities'. Page 4, Line 45 The paragraph beginning with 'previous reviews on coproduction in digital mental health interventions...' could be better expressed to synthesise the two publications cited. Page 5, line 9 The list of marginalised groups needs to be presented so that it is grammatically accurate. A major concern is the omission of indigenous young people from the list of marginalised young people. Further, you have stated earlier that your definition of young people who have disadvantage, vulnerable and or marginalised are those who come from a social, economic, cultural, or digital perspective. However, the list of marginalised groups does not refer to the digital perspective. You could include 'young people without access to the Internet'. page 5 line 30 If an exploratory approach is taken to the research question, why predefined the marginalised groups? It is possible that there are other marginalised groups that you have missed in this list that may be included in the literature. For example, other marginalised groups might include indigenous young people, young parents, young carers, young people whose parents have a mental illness or are in custody. Methods Page 5, line 53 The timeline for the review begins in October 2023. As it is now December 2023, are you open to changing the methods of this review, given that it seems the review may already be underway?
--	--

	Page 6, line 24 The period for publication of the articles seems to be very short, especially given that the Covid lockdowns may have limited the amount of research that has been undertaken, particularly research that used codesign of programs with young people. A longer period would potentially be more fruitful in answering your research question. Page 6, line 35 I wonder if you might consider presenting the search terms in a table so they can be digested more easily. I noticed that you are searching for six groups of articles and are then looking for the intersection of those groups (via the AND search function). Some articles may be missed due to the need to meet such a large number of groups of search criteria. I noticed that the Google Scholar search only requires the intersection of three groups, which might be more appropriate. page 7, line 24 In your inclusion criteria, have you considered including a percentage of the sample population that needs to be aged between 16 to 25? Will you include the study if it also includes children or adults? In the exclusion criteria, you state you will not include children under 16 and adults over 25, so what will you do, for example, if a study has a sample of young people aged 14 to 22? Page 7, line 30. Your eligibility criteria seem to have some elements overlapping in points three and four. Page 7, line 30 The age criteria being 16 to 25 may need some rationale given that many countries define the period of youth as beginning from 10 or 12 years old. Page 7 line 49 When charting the data, the data extraction fields include 'any other key findings'. Presumably, these are in relation to the research question. Also, I noticed that the template for charting does not exactly match the description in the text. For example, the text says you will include limitations. However, a field for limitations is missing from the template.
--	--

REVIEWER	Schick, Anita Heidelberg University
REVIEW RETURNED	29-Jan-2024

GENERAL COMMENTS	The manuscript is a protocol for a scoping review on co-production of digital mental health supports for young individuals from marginalized backgrounds. The protocol has been registered with the Open Science Framework prior to collecting the data, i.e. screening the literature. The present manuscript is more detailed than the registration and introduces the background and motivation for the scoping review. The manuscript is well-
--

	organized and well-written. It entails all details on methods in order to replicate the study. The topic of the scoping review is a rapidly evolving field and co-production may be one factor for successful implementation and maintenance of digital health apps. I only have a few minor comments to the authors:  - In some countries the development of digital health apps is regulated (e.g., in the EU by the Medical Device Regulation). It would therefore be of interest to include in the chart whether it was a medical product that was co-developed.
--	--

VERSION 1 – AUTHOR RESPONSE

Formatting Amendments (where applicable): Reviewer: 1 Ms. Fiona Robards, University of Sydney, University of Sydney Comments to the Author: Thank you for the opportunity to review 'coproduction of accessible digital mental health support in partnership with young people from marginalised backgrounds: a scoping review per protocol'. General comments The topic for this review is interesting and would provide a useful addition to the literature. I realise that youth is increasingly being used as a noun, such as in the expression marginalised youth. However, if you consult the dictionary, youth describes a period of time, not a group of people. I therefore advocate that youth is used as an adjective, not a noun. For example, 'marginalised young people' is much better than 'marginalised youth'.	We have addressed the use of the term youth and replaced it with young people where possible.
Abstract	A sentence has been added to the abstract to clearly articulate the research question.

The abstract reads well. My only suggestion is to see the research question clearly articulated within the abstract.	
Article summary The strengths and limitations section articulates that there may be a shortage of articles. I noticed that the search strategy calls for the intersection of six groups, and the requirement of articles meeting so many may limit the search. Further, the publication period is very limited. Can these limitations be addressed?	Using the initially proposed search terms, we have identified 20 studies for extraction/inclusion in the review. Using additional terms i.e. indigenous, refugee and neurodiver*, the search returned a further two studies, which we are now being considered for inclusion. This means there are a sufficient number of relevant studies available.
A strong limitation is the omission of indigenous young people in the list of marginalised groups and from the search terms. This issue should be addressed.	The title and abstract screening without the search terms indigenous resulted in the return of 8 publications (6 in Australia, 1 in New Zealand, and 1 in Canada) on indigenous young people for full text screening, with three (1 Australia, 1 in Canada, and 1 in New Zealand) being included in the final scoping review. Wanting to address the reviewer's concern of omission and having further discussions within the team, we decided to re-run our searches with the additional marginalisation terms indigenous, refugee, and neurodiver*. This resulted in two more publications (1 from Australia and 1 from the US - both indigenous young people) which we are considering for inclusion. We would also like to address this issue in terms of its cultural relevance in Ireland. The Irish indigenous population group, Travellers, is identified as a distinct ethnic minority group, and therefore, is covered by search terms used in this scoping review protocol.
Introduction Page 4, line 6 The first line of the introduction needs reconsideration. I suggest softening the language to include words such as 'some' and 'may' to make to improve the accuracy of the claim.	The language has been softened with the use of suggested words.

The first paragraph has both expressions 'with regards to', and 'with regard to'. I suggest editing for consistency.	For consistency, the expression has been changed to 'with regard to'.
Page 4, line 17 It would be good to define how digital mental health support is conceptualised. Does this include prevention through to treatment? Apps and online programs and telehealth?	We expanded on this point in the paragraph on page 4 "in the context of digital mental health supports, ranging from mental health promotion, and primary prevention to targeted interventions. This includes all types of digital interventions, such as websites and apps but not studies that report on digital delivery of services that are typically administered face-to-face such as remote counselling."
Page 4, line 31 While it is your choice, I would advise against using an acronym such as DVM. Later in the article (for example, in the research question), you refer to marginalised young people, which is much better.	We only used the acronym DVM to reflect the definition of marginalised young people provided in a briefing paper produced by UNICEF's Office of Research. However, we have now removed this abbreviation and use the term 'marginalised young people' throughout the paper.
Page 4, line 41 The example groups of marginalised young people must be grammatically accurate and described more respectfully. See the last sentence in the paragraph. It is not appropriate to refer to a group of young people as 'sexual minorities'.	The examples were taken from the briefing paper produced by UNICEF's Office of Research. The authors rephrased to address the reviewer's concern regarding the reference to sexual and gender minorities.
Page 4, Line 45 The paragraph beginning with 'previous reviews on coproduction in digital mental health interventions...' could be better expressed to synthesise the two publications cited.	The sentence has been rephrased to highlight the main argument both papers make in terms of marginalised young people.
Page 5, line 9 The list of marginalised groups needs to be presented so that it is grammatically accurate. A major concern is the omission of indigenous young people from the list of marginalised young people.	The omission of the search term 'indigenous young people' has been addressed in a previous response and has now been rectified by running additional searches.

Further, you have stated earlier that your definition of young people who have disadvantage, vulnerable and or marginalised are those who come from a social, economic, cultural, or digital perspective. However, the list of marginalised groups does not refer to the digital perspective. You could include 'young people without access to the Internet'.	In relation to the digital perspective in Ireland, living in isolated rural areas covers this aspect.
page 5 line 30 If an exploratory approach is taken to the research question, why predefined the marginalised groups? It is possible that there are other marginalised groups that you have missed in this list that may be included in the literature. For example, other marginalised groups might include indigenous young people, young parents, young carers, young people whose parents have a mental illness or are in custody.	Given the multitude of different groups, we focus on the Irish context when providing the list in bullet points. It is also repeatedly stated that the list of marginalised young people includes but is not limited to the number/types of groups presented. Searches with the initial search terms have returned papers relating to indigenous young people, young parents, and young people whose parents have mental health difficulties. To address concerns, we have conducted additional searches with the terms indigenous, refugee and neurodiver*. We feel that the initial, as well as now revised, search terms capture the diversity of marginalised groups and are thus adequate.
Methods Page 5, line 53 The timeline for the review begins in October 2023. As it is now December 2023, are you open to changing the methods of this review, given that it seems the review may already be underway?	We are happy to revise any sections of the protocol including re-running searches if deemed necessary to address concerns and allow for publication.
Page 6, line 24 The period for publication of the articles seems to be very short, especially given that the Covid lockdowns may have limited the amount of research that has been undertaken, particularly research that used codesign of programs with young people. A longer period would potentially be more fruitful in answering your research question.	We have provided a justification for the short timeframe in the manuscript: Highlighting an underrepresentation of marginalised young people, our searches will lead on from these reviews and be limited from January 2021 to the date of search commencement in October 2023, to ensure that the literature reflects the most current literature on the use of co-production methods and thus the nature of this dynamic research topic. Published work from 2021 onwards is included to mirror the current state of literature more accurately, given the accelerated use and assessment of digital mental health technologies throughout the sector during the pandemic (15).

Page 6, line 35 I wonder if you might consider presenting the search terms in a table so they can be digested more easily.	Search terms for academic databases are now presented in a table.
I noticed that you are searching for six groups of articles and are then looking for the intersection of those groups (via the AND search function). Some articles may be missed due to the need to meet such a large number of groups of search criteria. I noticed that the Google Scholar search only requires the intersection of three groups, which might be more appropriate.	The searches (including the additional search terms) have returned 2341 studies, with 1243 included for title and abstract screening and 73 for full text screening. We feel that this is a sufficient return to capture the literature for the timeframe.
page 7, line 24 In your inclusion criteria, have you considered including a percentage of the sample population that needs to be aged between 16 to 25? Will you include the study if it also includes children or adults? In the exclusion criteria, you state you will not include children under 16 and adults over 25, so what will you do, for example, if a study has a sample of young people aged 14 to 22?	We will include any study where the sample includes, but is not limited to, those aged 16-25. Where a study has a sample of e.g. young people aged 14-22, the authors will review the breakdown of age groups in relevant section.
Page 7, line 30. Your eligibility criteria seem to have some elements overlapping in points three and four.	Point 3 refers to co-production and point 4 relates to methodologies led by young people. Both points are included in order to capture these two distinct approaches.
Page 7, line 30 The age criteria being 16 to 25 may need some rationale given that many countries define the period of youth as beginning from 10 or 12 years old.	We have found that publications are conflicting in what age range constitutes youth/ young people. According to the UN, there is no universally agreed international definition of the youth age group. For statistical purposes, however, the United Nations—without prejudice to any other definitions made by Member States—defines 'youth' as those persons between the ages of 15 and 24 years. As we are keen to capture the out-of-school population of young people in Ireland, we opted to define young people as aged 16-25 to reflect the Education (Welfare) Act 2000. (Republic of Ireland) and the Education Reform (NI) Order 1989 Article 156 and DE Circular 1990/27 (Northern Ireland).

Page 7 line 49 When charting the data, the data extraction fields include 'any other key findings'. Presumably, these are in relation to the research question.	'Any other key findings' are in relation to the research question and this has now been amended in the manuscript.
Also, I noticed that the template for charting does not exactly match the description in the text. For example, the text says you will include limitations. However, a field for limitations is missing from the template.	The template refers to issues/debates identified. We have added an additional field for limitations.
Reviewer: 2 Dr. Anita Schick, Heidelberg University Comments to the Author: The manuscript is a protocol for a scoping review on co-production of digital mental health supports for young individuals from marginalized backgrounds. The protocol has been registered with the Open Science Framework prior to collecting the data, i.e. screening the literature. The present manuscript is more detailed than the registration and introduces the background and motivation for the scoping review. The manuscript is well-organized and well-written. It entails all details on methods in order to replicate the study. The topic of the scoping review is a rapidly evolving field and co-production may be one factor for successful implementation and maintenance of digital health apps. I only have a few minor comments to the authors:  - In some countries the development of digital health apps is regulated (e.g., in the EU by the Medical Device Regulation). It would therefore be of interest to include in the chart whether it was a medical product that was co-developed. 	The rationale for this scoping review is a focus on the co-design of digital mental health supports. The review is not concerned with the support being a medical device or with particular regulations around devices that were developed. As a result, we are not charting this information.

VERSION 2 – REVIEW

REVIEWER	Robards, Fiona University of Sydney, General Practice
REVIEW RETURNED	21-Mar-2024

GENERAL COMMENTS	Thank you for the opportunity to review 'Co-production of accessible digital mental health supports in partnership with young people from marginalised backgrounds: a scoping review protocol'. General comment The topic for this review would interest the journal readers and provide a useful addition to the literature. Abstract The abstract could be more succinct yet provide more detail on methods. Clarity about the scope of the literature (international, in English) should be included in the abstract. Article summary Page 2, line 13 The points on limitations acknowledge that there may be limited literature. If so, why include such a limited period (from 2021 onwards)? Introduction Page 4, line 11 List of marginalised groups. Does 'in care' also include 'in custody'? The list of marginalised groups needs to be grammatically correct. A major concern is that the list of marginalised groups are those "who have identified as marginalised in the Irish context". Why limit the marginalised groups to those that are relevant to Ireland? This limits the usefulness of the study to people in Ireland. It is a major concern that indigenous young people are being excluded from this study, which explores international literature on marginalised young people. Methods and analysis If the review is to be complete by Feb 2024, are you willing to take on changes identified by reviewers now in March 2024? Page 6 line 5 I noticed that some of the marginalised groups of interest are missing from the search terms. Page 6 line 50 The methods state that included articles are in English, but there is no mention of country. Therefore I conclude the literature must be international in scope, including low and high-income countries. The scope should be made much clearer in the methods and the abstract. Given that the scope is international, it is not appropriate to define marginalised groups are those only relevant to the Irish context. This is a major limitation. Page 7 line 10
--

	The eligibility criteria seem to have some elements overlapping in points three and four. Page 7 line 13 Young people are defined as aged 16-25. How will you manage the inclusion or exclusion of studies that have different age ranges (for example, 12 to 17 years)? The chosen age criteria (16-25 years) needs explanation, given that many countries define the period of youth as beginning from either 10 or 12 years old. Page 8 line 32 full stop typo. References Some are incomplete.
--	--

VERSION 2 – AUTHOR RESPONSE

Abstract

Reviewer comment: The abstract could be more succinct yet provide more detail on methods. Clarity about the scope of the literature (international, in English) should be included in the abstract.

Author response: We have included the scope (international and English) and tried to make abstract more succinct

Article summary

Reviewer comment: page 2, line 13. The points on limitations acknowledge that there may be limited literature. If so, why include such a limited period (from 2021 onwards)?

Author response: We have deleted our point regarding limitation, as we now know this is not the case.

Reviewer comment: Introduction. Page 4, line 11. List of marginalised groups. Does 'in care' also include 'in custody'? The list of marginalised groups needs to be grammatically correct.

Author response: No, 'in care ' does not include 'in custody'. We have removed the term in the listing now as it has not been included in the search terms.

Reviewer comment: A major concern is that the list of marginalised groups are those "who have identified as marginalised in the Irish context". Why limit the marginalised groups to those that are relevant to Ireland? This limits the usefulness of the study to people in Ireland. It is a major concern that indigenous young people are being excluded from this study, which explores international literature on marginalised young people.

Author response: The list on page 4 only provides the Irish context. We do state "the scoping review will include, but not be limited to, the following groups of young people". However, it is also important

to note that the search returned papers from various countries, most commonly USA but also Australia, Germany, UK, Canada, India, Africa (etc). Thus the terms cover marginalisation in a broad sense, but include some (e.g. travellers) that are specific to, but not limited to an Irish context. Traveller's (in Ireland the UK) are an example of a marginalised group similar to others elsewhere (e.g. Romany/Romani in Spain, Romania and Turkey).

Methods and analysis

Reviewer comment: If the review is to be complete by Feb 2024, are you willing to take on changes identified by reviewers now in March 2024?

Author response: We are happy to make any reasonable revisions to the protocol to improve the paper for publication, and therefore have paused progress on the scoping review at present. However, the authors are eager to move on to the write up of the scoping review as soon as possible once the reviewer's concerns have been addressed.

Reviewer comment: Page 6 line 5. I noticed that some of the marginalised groups of interest are missing from the search terms.

Author response: We have removed 'in care' now from the list of page 4 and believe that search terms presented on page 6 consist of definition terms provided by UNICEF's briefing paper as well as search terms found adequate to capture the groups of young people identified as marginalized in the international and Irish context.

Reviewer comment: Page 6 line 50. The methods state that included articles are in English, but there is no mention of country. Therefore I conclude the literature must be international in scope, including low and high-income countries. The scope should be made much clearer in the methods and the abstract. Given that the scope is international, it is not appropriate to define marginalised groups are those only relevant to the Irish context. This is a major limitation.

Author response: We have added a sentence to reflect international literature in English and feel that our search terms reflect the international and Irish context.

Reviewer comment: Page 7 line 10. The eligibility criteria seem to have some elements overlapping in points three and four.

Author response: We addressed this comment in R1: Point 3 refers to co-production and point 4 relates to methodologies led by young people. Both points are included in order to capture these two distinct approaches.

Reviewer comment: Page 7 line 13. Young people are defined as aged 16-25. How will you manage the inclusion or exclusion of studies that have different age ranges (for example, 12 to 17 years)? The chosen age criteria (16-25 years) needs explanation, given that many countries define the period of youth as beginning from either 10 or 12 years old.

Author response: We addressed this comment in our first revision: We will include any study where the sample includes, but is not limited to, those aged 16-25. Where a study has a sample of e.g. young people aged 14-22, the authors will review the breakdown of age groups in relevant section. Additionally, we added a note within text to explain the choice of age group 16-25.

We have found that publications are conflicting in what age range constitutes youth/ young people. According to the UN, there is no universally agreed international definition of the youth age group. For statistical purposes, however, the United Nations—without prejudice to any other definitions made by Member States—defines 'youth' as those persons between the ages of 15 and 24 years. As we are

keen to capture the out-of-school population of young people in Ireland, we opted to define young people as aged 16-25 to reflect the Education (Welfare) Act 2000. (Republic of Ireland) and the Education Reform (NI) Order 1989 Article 156 and DE Circular 1990/27 (Northern Ireland).

Reviewer comment: Page 8 line 32. full stop typo.

Author response: Full stop inserted after supports on the second last sentence of the discussion

Reviewer comment: References. Some are incomplete

Author response: The references have now been updated and missing information added.